# Dataset for Detecting the Electrical Behavior of Photovoltaic Panels from RGB Images

Juan-Pablo Villegas-Ceballos [1,*] , Mateo Rico-Garcia [2] and Carlos Andres Ramos-Paja [3]

1    Departamento de Electrónica y Telecomunicaciones, Instituto Tecnológico Metropolitano,
     Medellín 050015, Colombia
2    Facultad de Ingeniería, Institución Universitaria Pascual Bravo, Calle 73 No. 73A 226,
     Medellín 050034, Colombia; mateo.rico@pascualbravo.edu.co
3    Facultad de Minas, Universidad Nacional de Colombia, Carrera 80 No. 65-223, Medellín 050041, Colombia;
     caramosp@unal.edu.co
*    Correspondence: juanvillegas@itm.edu.co

**Abstract:** The dynamic reconfiguration and maximum power point tracking in large-scale photovoltaic (PV) systems require a large number of voltage and current sensors. In particular, the reconfiguration process requires a pair of voltage/current sensors for each panel, which introduces costs, increases size and reduces the reliability of the installation. A suitable solution for reducing the number of sensors is to adopt image-based solutions to estimate the electrical characteristics of the PV panels, but the lack of reliable data with large diversity of irradiance and shading conditions is a major problem in this topic. Therefore, this paper presents a dataset correlating RGB images and electrical data of PV panels with different irradiance and shading conditions; moreover, the dataset also provides complementary weather data and additional image characteristics to support the training of estimation models. In particular, the dataset was designed to support the design of image-based estimators of electrical data, which could be used to replace large arrays of sensors. The dataset was captured during 70 days distributed between 2020 and 2021, generating 5211 images and registers. The paper also describes the measurement platform used to collect the data, which will help to replicate the experiments in different geographical locations.

**Keywords:** photovoltaic; image-based estimation; partial shading; current vs. voltage characteristic



## 1. Introduction

In 2020, the power generation capacity based on photovoltaic (PV) installations grew by 130 GW, which includes both on-grid and off-grid solutions, reaching a total of 760 GW worldwide [1]. Moreover, PV systems are under continuous development to improve both efficiency and reliability, where the main focus has been placed on the PV array configuration, control systems and power converters, among others [2].

The partial shading is one of the most detrimental phenomena over PV systems, since such a phenomenon produces a large reduction in the power production, an increment in the panels' temperature, which reduces the system lifetime, and even could produce hotspots that damage the PV source [3,4]. This detrimental phenomenon has been addressed by introducing bypass diodes on the back of the PV panels [5], but due to the negative operation voltage of the diodes, such a solution allows the panels to consume power, thus degrading the PV cells and reducing the system lifetime. Another solution is based on dynamically changing the electrical connection of the PV panels to avoid the activation of bypass diodes [5], but such a solution requires a large number of voltage and current sensors

(one for each panel), and a large number of analog-to-digital converters (ADC) in order to acquire such current vs. voltage (I-V) data, which are needed to process optimization algorithms for detecting a safe connection for the PV panels [6].

With the aim of reducing the number of sensors and ADC, which in turn can reduce the system's complexity, size and cost and increase the system's reliability, some authors have proposed image-based methods to reconfigure PV systems. For example, in [7], the authors use a video camera to detect shades covering PV arrays. In particular, this method segments the images using morphological operations and thresholds on the HSV space, which is used to determine if the shade is a transient or constant condition for a given interval of time; finally, such information is used to reconfigure the PV array. Similarly, in [8], the authors propose a fuzzy logic solution to determine the optimal array configuration; this solution is based on image recognition to detect shades affecting the PV array. The authors of [9] report a slightly different approach, where an algorithm assigns weighted values to the pixels of an image, and such values are used to estimate the power degradation in each PV panel due to the shades. Finally, this information is used to estimate the configuration producing the highest power for dynamic reconfiguration purposes.

Another approach was reported in [10], which uses images to determine, indirectly, the irradiance reaching the panels. This solution uses a Lambertian surface to determine the overall irradiance on the installation site from the images, which is extrapolated to the whole PV source. Instead, the approach reported in [11] proposes a tracking shade method based on optic flux, which requires more computational effort. It is worth noting that the automatic detection of PV sources based on aerial images has been extensively studied [12,13], mainly using public data [14]. However, this task could also be performed using ground-level images and neural networks, which have demonstrated a satisfactory result in object segmentation applications [15].

The previous revision of the state-of-the-art shows the advantages of developing image-based reconfiguration systems, where multiple authors have addressed the shade detection problem using fuzzy or evolutionary techniques, which must be trained and tested using real data formed by both RGB images and electrical I-V data. However, obtaining images and electrical data for a large amount of different shading and irradiance conditions requires a particular infrastructure and a long experimental campaign. Image-based datasets for PV systems have been reported in the literature for other applications; thus, the characteristics of the data are not suitable for training models for predicting the I-V data from images. For example, the work reported in [16] provides thermal images oriented to detect hot-spots in the PV array; thus, no electrical data are provided. Similarly, the dataset reported in [14] is oriented to detect PV installations using satellite images, and not electrical data. Other datasets provide electrical data, but with different objectives. This is the case of the dataset reported in [17], which relates the energy production to panels' parameters, such as aging, while [18] relates the energy production to the installation site.

Therefore, this work improves the state-of-the-art by providing datasets formed by the RGB images of the panels, the I-V data and the power vs. voltage (P-V) data, the fill factor and complementary weather data such as irradiance, temperature, Azimuth angle, Zenith angle and Albedo. Moreover, the dataset also provides additional image characteristics such as the blue channel of the RGB space, the luminance histogram (Y) of the Y-Cb-Cr space, the saturation histogram from the HSV space, the histogram of the third channel from the NTSC space and the histogram of the first channel from the CIE 1976 space. This dataset is oriented to support the training of I-V curve estimation models for developing image-based reconfiguration systems, where a single RGB camera could replace a large number of current and voltage sensors. In addition to the dataset, the hardware setup used to collect the data is also described, which could be used to replicate the experimental campaign in other geographical locations.

The paper is organized as follows. Section 2 describes the data acquisition system and the procedures used to collect the dataset, and Section 3 describes the data structure and provides some examples. Then, Section 4 provides an analysis of the data, which shows the

applicability of the dataset for detecting the short-circuit current and maximum power from the images; the short-circuit current is commonly used in reconfiguration systems, while the maximum power is commonly used in MPPT techniques. Finally, Section 5 presents the conclusions, and the dataset can be accessed from [19].

## 2. Materials and Methods

The block diagram of the data acquisition system is presented in Figure 1a. This setup uses an electronic load BK8514 from B&K Precision to impose a voltage sweep from open-circuit voltage ($V_{oc}$) to 0 V at the PV panel terminals, which produces a current sweep from 0 A to short-circuit current ($I_{sc}$). The panel current and power are measured by the electronic load, and a Python algorithm acquires both I-V and P-V data using a USB connection between the electronic load and a computer. On the other end, the images of the panel surface are recorded using a DS-2CE16C2T-IR camera from Hikvision, which is normally used for surveillance. This camera captures a minimum illumination of 0.01 Lux, and it has the Automatic Gain Control (AGC) turned on. The Python algorithm acquires the image using an Ethernet connection between the camera and the computer. The connection between the PV panel and the electronic load is performed with two cables 2pfg-1169 PV1-F, which have a section of 4 mm$^2$, introducing a resistivity of 4.97 $\Omega$/km. Since the cables have a longitude of 4 m, the voltage measurement will need a correction of 19.88 m $\Omega \times i_{pv}$, where $i_{pv}$ is the panel current. Figure 1b shows the physical setup of the data acquisition system installed outside.

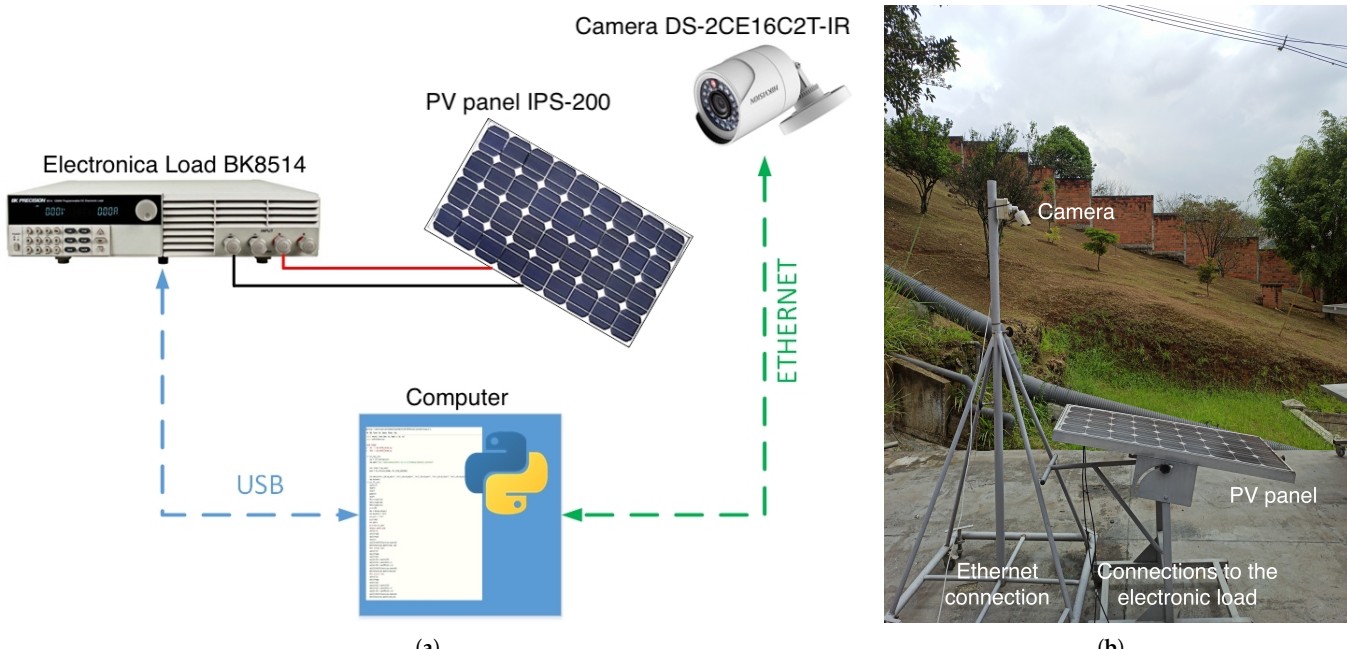

**Figure 1.** Data acquisition system. (**a**) Block diagram. (**b**) Physical setup.

The PV source utilized in this study is a BP585 panel, which has a maximum power ($P_{mpp}$) of 85 W; all the electrical characteristics of such a panel are reported in Table 1, where $AM$ is the mass of the air, $E$ is the solar irradiance, and $T_p$ is the panel temperature. In the experiments, the PV panel was oriented to the south with a tilt angle of 10°.

**Table 1.** PV characteristics

| Characteristics | Values |
|---|---|
| Panel Dimension (H/W/D) | $1204 \times 538.5 \times 50$ mm |
| Maximum Power ($P_{max}$) | 85 W |
| Current at Maximum Power ($I_{mpp}$) | 4.83 A |
| Voltage at Maximum Power ($V_{mpp}$) | 18.02 V |
| Short-Circuit Current ($I_{sc}$) | 5.22 A |
| Open-Circuit Voltage ($V_{oc}$) | 21.85 V |
| Temperature Coefficient of $I_{sc}$ | $(0.065 \pm 0.015)\%/°C$ |
| Temperature Coefficient of Voltage | $-(80 \pm 10)$ mV/°C |
| Temperature Coefficient of Power | $-(0.5 \pm 0.05)\%/°C$ |
| Nominal Operating Cell Temperature (NOCT) | $47 \pm 2$ °C |
| Fill Factor (*FF*) | 0.98 |
| Standard Test Conditions (*STC*) | AM = 1.5, |
|  | E = 1000 W/m$^2$, |
|  | $T_p = 25$ °C |

The measurements were taken on the Campus Fraternity of the Intituto Tecnologico Metropolitano (ITM), which is located on Medellin–Colombia at the coordinates $6°14'39.9''$ N latitude and $75°33'07.1''$ W longitude; the geographical location can be observed in Figure 2a. The experimental setup was located on the side of the building, as observed in Figure 2b, which provides multiple shading profiles throughout the day. Therefore, such a location enables the production of a dataset with multiple shading conditions. The experimental campaign was conducted in two intervals to account for different seasonal conditions: first from 17 March 2020 to 2 May 2020, and secondly from 22 September 2021 to 14 October 2021. Finally, such an experimental campaign generated 5211 datasets, each one of them formed by an image and the associated electrical I-V and P-V characteristics. Moreover, the dataset also provides both weather conditions and image characteristics for each register.

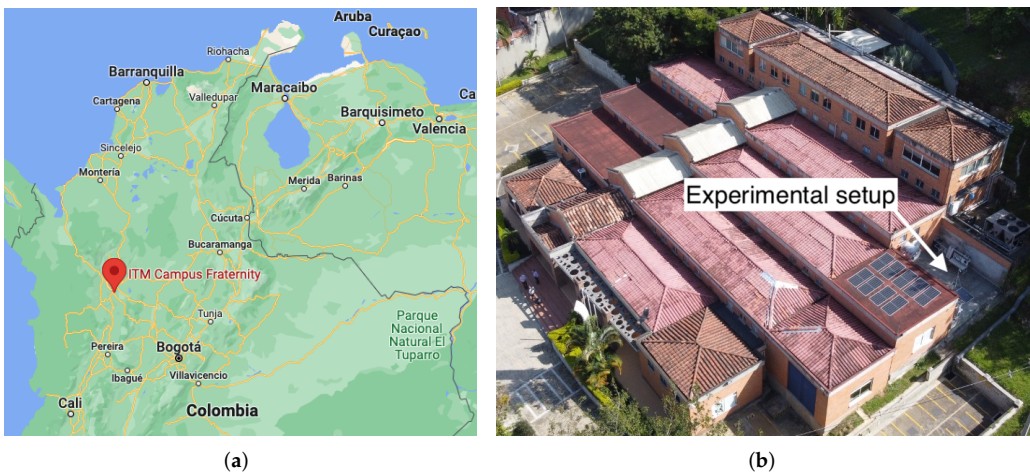

(a)  (b)

**Figure 2.** Location of the experiments. (**a**) Geographical location. (**b**) Physical location.

The production and acquisition of the dataset was programmed on a Python algorithm, which managed both the camera and electronic load. The process pseudocode is reported in Algorithm 1, which enables its implementation in any programming language. The dataset is formed by data samples taken in intervals of 5 min, starting from 6:00 to 18:00.

---

**Algorithm 1** Data production and acquisition.

---

Camera initialization
Electronic load initialization
Set sample time
Set sweep data: Vmin, Vmax, Imin, Imax
**while** Camera available and electronic load available **do**
  **if** Sample time **then**
    Acquire and save image
    **for** Voltage sweep points **do**
      Set voltage to the electronic load
      Save voltage
      Acquire and save current and power
    **end for**
  **end if**
**end while**

---

The voltage sweep for each operation condition was performed from open-circuit voltage (24 V) in intervals of 0.5 V each 200 ms, producing a total of 47 measurements for each image, and requiring 14.1 s to complete the voltage sweep. This sweep speed is fast enough to avoid large irradiance and shadow changes, but also slow enough to avoid sensible errors due to the internal capacitance of the PV panel. Finally, the flowchart reported in Figure 3 summarizes the data production and acquisition process.

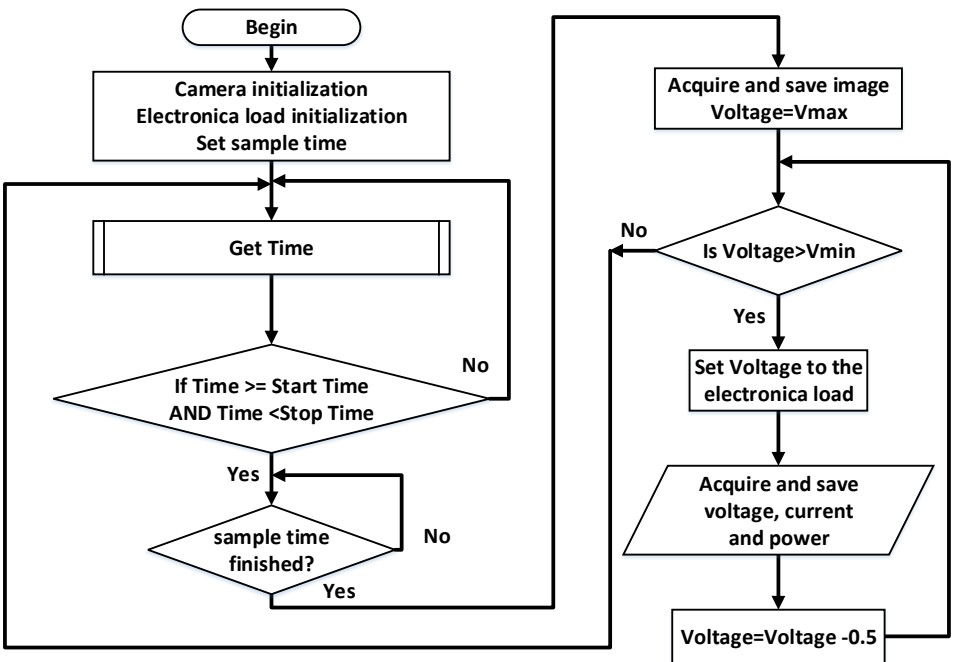

**Figure 3.** Flowchart of the data production and acquisition process.

In addition to the current, voltage and power, another factor widely adopted to analyze the panel efficiency is the fill factor (*FF*), given in (1), which provides a relation between the maximum power and the product between the short-circuit current and open-circuit voltage.

$$FF = \frac{P_{mpp}}{V_{oc} \times I_{sc}} \tag{1}$$

An example of this approach is reported in [20], where the *FF* is used to compare the efficiency of different PV array configurations under the same partial shading condition;

thus, a higher *FF* implies the higher efficiency of the array configuration for this particular shade. Moreover, the *FF* is also used to detect panel degradation. For example, the work reported in [21] defines the normalized fill factor (*NFF*) given in (2) to detect degradation, where the irradiance is estimated from the short-circuit current, and $V_{oc,ST}$ and $I_{sc,ST}$ are the open-circuit voltage and short-circuit current in STC conditions given in Table 1 (manufacturer or datasheet data). In the mentioned work, it was demonstrated that decrements in *NFF*, for the same short-circuit current, correspond to panel degradation.

$$NFF = \frac{FF}{V_{oc,ST} \times I_{sc,ST}} \tag{2}$$

Therefore, the *FF* was included into the dataset; thus, the data could be used to analyze the panel efficiency and degradation. Moreover, the data could also be used to train models to detect the *FF* from the RGB images with shading conditions. Finally, the *NFF* can be easily calculated from the *FF* and the STC data given in Table 1; thus, the *NFF* was not included in the data to avoid an increment in the dataset size.

The images and electrical data are related by the file names: both files have the same name, which have the structure year_day_month_hour_minute ("YYYY_DD_MM_HH_MM"). The images have a JPG extension, while the electrical data are recorded into text files with three columns (voltage, current and power of the panel). Moreover, the images were processed to provide a better perspective of the PV panel, and these processed images were also added to the dataset. Such image processing proceeded as follows: the PV panel was cropped to remove information from the place where the experiment was conducted, and the image perspective was corrected to show the PV panel in a portrait view. These additional images will be useful to train algorithms requiring exclusively the panel image, while the unprocessed images will be useful to train algorithms with mobile cameras, e.g., drone-mounted devices. Finally, the processed images can be an additional validation set.

The dataset also includes a single file (named Features.cvs) with important characteristics of each experiment. In this file are reported the $V_{oc}$, $I_{sc}$, $P_{mpp}$ and *FF* for each image, which simplifies the data processing for training models for these parameters. Moreover, this file also reports weather data for each image: irradiance in W/m$^2$, temperature in °C, Azimuth angle, Zenith angle and Albedo.

With the aim of supporting the processing of the images from the dataset, some data extracted from each image were also included into the Features file. Such additional characteristics were selected based on the traditional requirements of image processing algorithms:

- The blue channel of the RGB space;
- The luminance histogram (Y) of the Y-Cb-Cr space;
- The saturation histogram from the HSV space;
- The histogram of the third channel from the NTSC space;
- The histogram of the first channel from the CIE 1976 space.

In fact, the previous characteristics are used in Section 4 to illustrate the usability of the dataset, and the selection of these characteristics is discussed. Finally, the structure of the dataset is the following one:

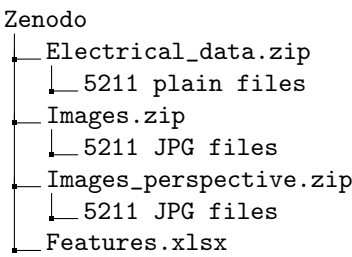

```
Zenodo
├── Electrical_data.zip
│   └── 5211 plain files
├── Images.zip
│   └── 5211 JPG files
├── Images_perspective.zip
│   └── 5211 JPG files
└── Features.xlsx
```

### 3. Dataset Examples

This section presents some examples of the dataset. Figure 4 shows some of the images from the dataset; the first one (Figure 4a) was captured at 15:59 of 01 May 2020; the second one (Figure 4b) at 08:25 of the same date, 02 May 2020; the third one (Figure 4c) at 06:03 of 08 June 2020, and the fourth one (Figure 4d) at 15:44 of 24 September 2021. These figures exhibit different tones caused by shades covering part or all of the panel.

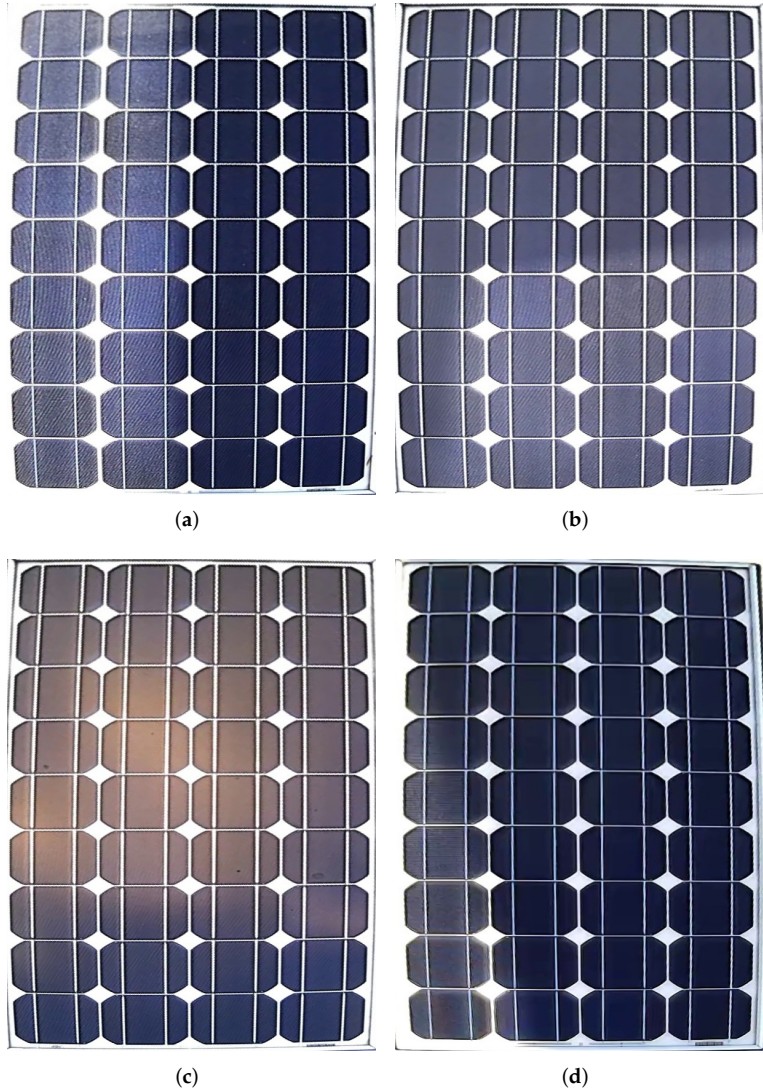

**Figure 4.** Four examples of the images in the dataset. (**a**) 2020_01_05_15_59.jpg. (**b**) 2020_02_05_08_25.jpg. (**c**) 2020_08_04_06_03.jpg. (**d**) 2021_24_09_15_44.jpg.

Figure 5 reports the P-V and I-V electrical data corresponding to the same panel figures, and these electrical data are recorded in the dataset (as text files) with the name associated with each figure. The data show that the changes in the irradiance and shading conditions impose different maximum power points to the panel (11.62 W, 15.41 W, 0.34 W and 5.96 W), and even a multi-peak condition is observed for the data collected at 15:59 of the 01 May 2020. Moreover, from such electrical data, it is possible to extract $V_{mpp}$, $I_{mpp}$, $P_{mpp}$ $V_{oc}$ and $I_{sc}$ for all the operation conditions in the dataset.

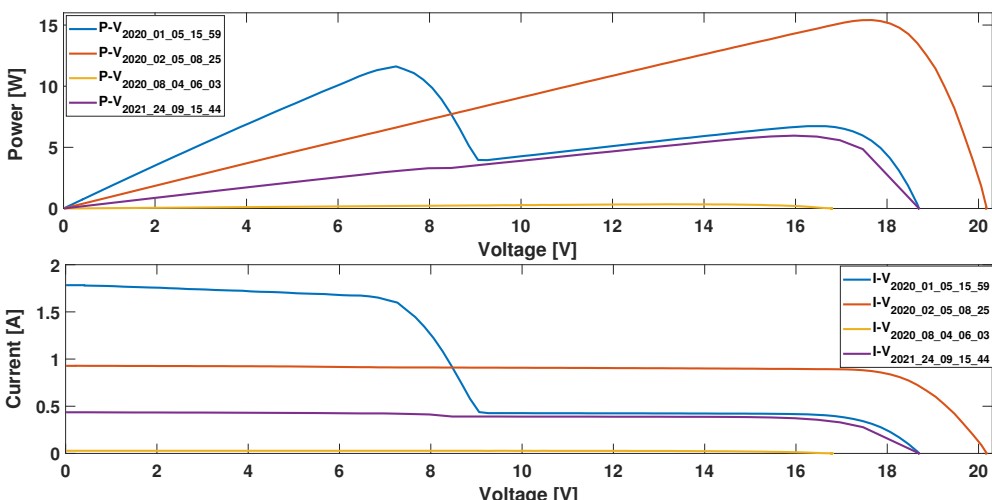

**Figure 5.** P-V and I-V curves for four samples from database.

An additional analysis is performed to illustrate the usability of the fill factor. For the data collected on 2 May 2020, the *FF* is 0.83, while the *FF* values obtained for 8 June 2020 and 24 September 2021 are 0.80 and 0.82, respectively. It is observed that the panel efficiency is similar for these conditions despite the difference in the power production, which is expected since the data were collected on close dates; thus, no significant degradation occurred between measurements; moreover, these data do not exhibit partial shading conditions. Instead, the data collected on 1 May 2020 have a partial shading condition, producing a *FF* = 0.38, which shows the efficiency reduction of the PV panel caused by such a partial shading. The result of this type of analysis can be used to model the efficiency impact of partial shading conditions on a larger PV array.

## 4. Example: Prediction of $I_{sc}$ and $P_{mpp}$ from the Images Using a Standard Algorithm

The main intention of the dataset is to support the prediction of electrical data from images, which is useful for reconfiguration systems, MPPT solutions and even for diagnostic purposes. Therefore, this section evaluates the applicability of the dataset to this problem; in particular, the estimation of the short-circuit current $I_{sc}$ and maximum power $P_{mpp}$ is evaluated using a standard regression model. For this example, the processed images of the dataset were used, since these images only show the PV panel in portrait perspective, which simplifies the extraction of color characteristics. In this way, color characteristics in different spaces were selected to train the model, with the objective of detecting the difference between the incident light from the images. The characteristics selected for the prediction model examples are: the blue channel of the RGB space, the luminance histogram (Y) of the Y-Cb-Cr space, the saturation histogram from the HSV space, the histogram of the third channel from the NTSC space and the histogram of the first channel from the CIE 1976 space. These characteristics were selected based on previous works [22], but taking care that each characteristic contributed useful information to enable the differentiation between the images. The model training was performed with the Regression Learner App from Matlab, which enables the evaluation of different types of algorithms. From these tests, the Rational Quadratic Gaussian Process Regression (GPR) algorithm was selected.

The training of both models (one for $I_{sc}$ and another for $P_{mpp}$) was performed using cross-validation with 5 folds, where the GPR algorithm with constant basis function was selected. The prediction models used the following training parameters, obtained from trial and error: kernel scale = 1000.2469, signal standard deviation = 14.0142 and automatic sigma. Figure 6 shows the prediction results for $P_{mpp}$, which exhibit a root mean square error (RMSE) equal to 7.96 and a training time of 439.38 s. Therefore, these results evidence the applicability of the dataset to train prediction models for the maximum power.

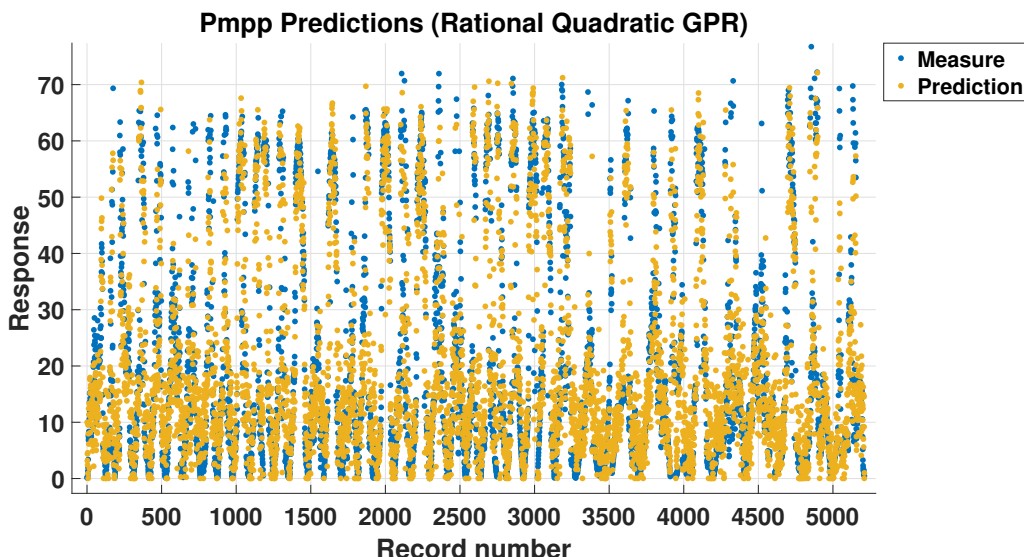

**Figure 6.** Maximum power predictions using the Bagged Trees method.

Figure 7 shows the prediction results for the $I_{sc}$ model, which is characterized by an RMSE = 0.6101 and a training time equal to 452.1 s. Similar to the previous case, these results confirm the applicability of the dataset to train prediction models for the short-circuit current.

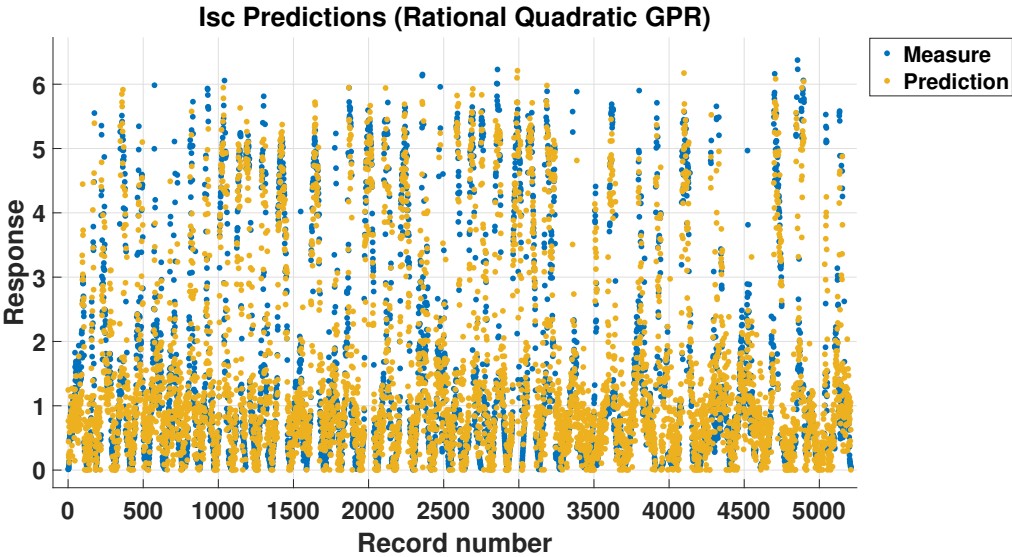

**Figure 7.** Short-circuit current predictions using the Bagged Trees method.

Figure 8 reports the prediction errors of both $P_{mpp}$ and $I_{sc}$ models for each year of the experimental campaign. The prediction of $P_{mpp}$ for 2020 and 2021 has average errors of 22.53%, where the predictions exhibit errors below 42.03% for 75% of the 2020 data, and errors lower than 9.38% for 25% of the same data. For 2021, the errors were below 32.5% for 75% of the data, and below 7.63% for 25% of the data. The prediction of $I_{sc}$ has an average error of 20.96% for the 2020 data, with errors below 38.48% for the 75% of the data and below 9.04% for 25% of the data. For 2021, the average error was 16.16%, with errors below 29.94% for the 75% of the data and below 7.22% for 25% of the data.

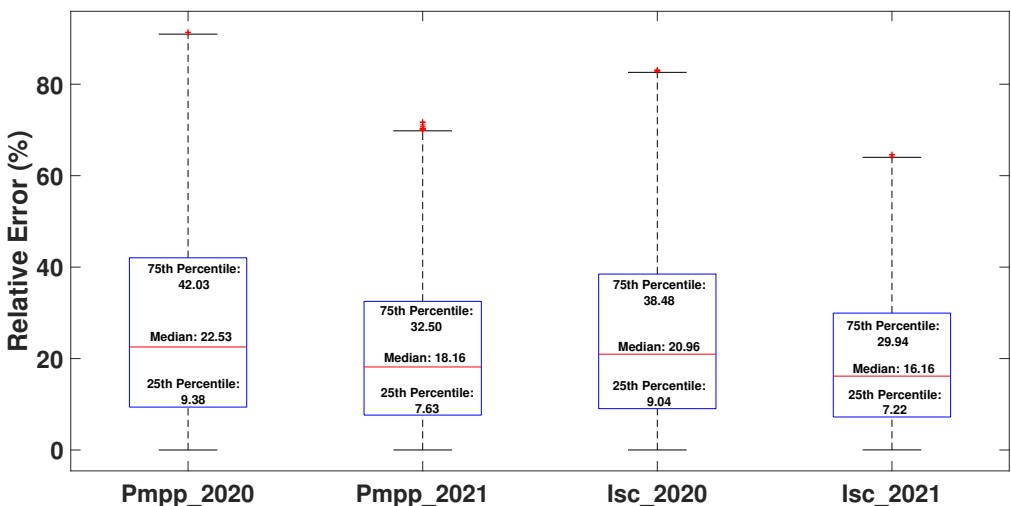

**Figure 8.** Relative error distribution by year for $P_{mpp}$ and $I_{sc}$ estimation.

It must be noted that the previous example is aimed at illustrating the usability of the dataset by training a standard Matlab model. Therefore, the results can be improved by adopting more efficient solutions. Moreover, additional characteristics can be taken into account to improve the information extracted from the images. Finally, the additional weather data were not used in this example; thus, including some of these data could lead to improved predictions. In any case, this example shows the methodology to train prediction models for the estimation of electrical characteristics from PV panel images.

## 5. Conclusions

This paper has presented both a dataset and measurement platform for RGB images and electrical data for a PV panel with different irradiance and shading conditions. Such a dataset was designed for training prediction systems to estimate electrical data from images, which could be used to replace costly and bulky sensors on reconfiguration systems. Moreover, this dataset can be also used to develop maximum power point tracking (MPPT) techniques based on images for partially shaded PV arrays, which could avoid the installation of large amounts of voltage and current sensors.

The dataset provides a large array of environmental data, which could also be useful to develop prediction models including temperature and irradiance, among other measurements. However, the panel temperature is not provided, which introduces an uncertainty factor caused by the variation in the temperature for the same irradiance condition. In any case, the panel temperature can be estimated from the ambient temperature, NOCT and irradiance value, as reported in [21]. The estimation of the panel temperature can be further improved by using the method reported in [23], which also enables us to estimate the temperature at the back of the module, thus providing higher precision in comparison with the model reported in [21], but requiring additional measurements such as wind speed.

The dataset is publicly available, and the data are very simple to use since the images and electrical data are linked by the file names. In addition, a preliminary analysis confirmed the usability of the data to train prediction models for the short-circuit current and maximum power, but models to predict any other characteristic can be designed, e.g., open-circuit voltage, fill factor, voltage of the bypass diode activation, etc. Moreover, the images can be divided to relate the light intensity of a particular cell with the power production of the whole panel, which can be useful for reconfiguration purposes or partial shading analysis. However, the electrical data of the individual cells are not provided; this could be a useful new experimental campaign, but it will require a much more complex experimental setup to access all the cell terminals.

Finally, the dataset is also useful for efficiency analyses and the study of the partial shading effect. For this purpose, the PV power that could be produced without partial shading conditions, for a particular irradiance condition, can be estimated from the single-diode model, whose parameters can be extracted using the method reported in [21].

**Author Contributions:** Conceptualization, J.-P.V.-C., M.R.-G. and C.A.R.-P.; methodology, J.-P.V.-C. and M.R.-G.; software, J.-P.V.-C. and M.R.-G.; validation, J.-P.V.-C., M.R.-G. and C.A.R.-P.; formal analysis, J.-P.V.-C., M.R.-G. and C.A.R.-P.; investigation, J.-P.V.-C., M.R.-G. and C.A.R.-P.; data curation, J.-P.V.-C. and M.R.-G.; writing, J.-P.V.-C., M.R.-G. and C.A.R.-P.; visualization, J.-P.V.-C. and M.R.-G. All authors have read and agreed to the published version of the manuscript.

**Funding:** This research work has been funded by the Institución Universitaria Pascual Bravo, Instituto Tecnológico Metropolitano and Universidad Nacional de Colombia under the project "Reconfiguración de paneles fotovoltaicos para la maximización de extracción de potencia bajo condiciones de sombreado basado en procesamiento de imágenes con inteligencia artificial", with codes IN202102, PE21105 and Hermes 51503, respectively. The APC has been funded by Instituto Tecnológico Metropolitano and Universidad Nacional de Colombia.

**Institutional Review Board Statement:** Not applicable.

**Informed Consent Statement:** Not applicable.

**Data Availability Statement:** The dataset is available from https://doi.org/10.5281/zenodo.6386767, accessed on 8 April 2022.

**Acknowledgments:** The authors thank the Institución Universitaria Pascual Bravo, Instituto Tecnológico Metropolitano and Universidad Nacional de Colombia. Moreover, the authors thank Solcast for supporting this work by providing valuable irradiance and weather data.

**Conflicts of Interest:** The authors declare no conflict of interest.

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
