# Peer review of "Dataset for Detecting the Electrical Behavior of Photovoltaic Panels from RGB Images"

_data, 2022_

Round 1

Reviewer 1 Report

The paper “Dataset for detecting the electrical behaviour of photovoltaic panels from RGB images” presents a dataset formed by the RGB images of the panels, the I-V data and the power vs. voltage (P-V) data with different irradiance and shading conditions. 
This topic could be of interest to the readers of this journal.

Specific comments

Abstract
Specific and quantitative results and findings have to be included.

1. Introduction 
The novelty of the work and where does it go beyond previous efforts in the literature have to be remarked. 
“ This dataset can be also used to develop maximum power point tracking (MPPT) techniques based on images for partially shaded PV arrays,  which could avoid the installation of a large amount of voltage and current sensors”
Since this topic is not investigated in the current research, this statement can be added to the conclusions as to future developments.

2. Data acquisition system
This section has to be enriched and it might be renamed “material and methods”.
Effect on power and efficiency reduction due to shading and irradiance effects have to be included.
Table 1. The PV temperature coefficient and efficiency have to be added.
“Tc is the cell temperature”  please check!
Some references should be added (e.g. Multilayer thermal model for evaluating the performances of monofacial and bifacial photovoltaic modules IEEE Journal of Photovoltaics 10 (4), 1035-1043)
Azimuth and tilt angle of the PV modules have to be added
Algorithm 1- Please add an illustrative  flow-chart 

3. Dataset examples
Figure 3. the shadowed surface is not evidenced. Please provide more info on this.
What were the shadowed cells?
Figure 4. How did the curves were  obtained
What were the Plane of Array (POA) Irradiances?  What were the expected nominal powers?
Prediction of electrical data from the images
“The characteristics selected for the prediction models  are:…………..”

The methodology used for elaborating the images as well as The training of both models (one for Isc and another for Pmp have to be further illustrated in the suggested material and methods section.

Please add further comments for  diagnostic purposes

Author Response

We thank the editor and reviewers for their comments aimed at improving the quality of our paper Dataset for detecting the electrical behavior of photovoltaic panels from RGB images (data-1635096).

In the following we address their concerns point by point.

Reviewer 2 Report

The dataset could be useful, however due to following concerns its leveraging in evaluating the pv systems performance could be difficult.

-it is of for single location, additionally there is no validation of the data.

-Monitoring setup details are not given in details.

-RGB images seems missing?

-The characteristics results data point should be provided.

-Additionally please show the data validation, how one should use these data's, why should they use when there are other data's existing in the form the NASA, NREL, JRC databases.

Author Response

(The authors gave the same response as above.)

Reviewer 3 Report

This article addresses an important topic in PV panel operation namely management of partial shading conditions.
After reading and analyzing the article, I can formulate the following observations:

- The dataset does not contain temperature. In similar irradiation conditions, Isc and Pmax can differ if temperature is different. In case of 0.5%/C degree temperature coefficient for Pmax (typical value for most panels), 10 degrees can leads to 5% difference. This issue must be stated as possible limitation of the dataset;

- Does the video camera has AGC (Automatic Gain Control) deactivated during recordings? (it is important because AGC make camera less sensitive to image brightness variations -> can alter information related to actual irradiation level);

- Did panel voltage measurements were performed using separate sensing wires? If not, even a 10 m solar cable can introduce about 0.4V voltage drop (at 8A) that alter voltage measurements. In this case, authors must specify cable characteristics (especially resistance) in order to make voltage drop compensation possible;

- What is the duration of the voltage sweep? If it was very short (less than 10 milliseconds) it is possible that internal capacitance of the solar cell to generate errors in current measurement;

- How the characteristics selected for the prediction models were extracted? For the whole image of the panel or the panel image was divided in several sections (possibly down to cells level) in order to evaluate how shading affect cells independently?

- Figures 5 and 6 are difficult to interpret. It would be better to represent prediction error instead of actual/predicted values.

- What if time of captured image would be excluded from training data? I suspect that time can get more information about shadow shape than image itself (sun position is time dependent therefore shadow placement of PV panel can be predicted from time information). It would be interesting to evaluate Isc and Pmax prediction only from image information.

In conclusion, I recommend to reconsider the article after major revision in order to address the observations presented above. 

Author Response

(The authors gave the same response as above.)

Round 2

Reviewer 1 Report

The Authors have improved their paper following the reviewer's comments.

Reviewer 2 Report

Authors have responded clearly. The manuscript is improved and can be accepted.

Reviewer 3 Report

I read the revised version of the article and I noticed that the main observations were properly addressed (either by adding/modify the paper or by giving explanatory comments in the author response document).

In conclusion, I recommend to accept the article in the present form.